# Incorporating Biomarkers in COPD Management: The Research Keeps Going

**DOI:** 10.3390/jpm12030379

**Published:** 2022-03-01

**Authors:** Ioannis Pantazopoulos, Kalliopi Magounaki, Ourania Kotsiou, Erasmia Rouka, Fotis Perlikos, Sotirios Kakavas, Konstantinos Gourgoulianis

**Affiliations:** 1Department of Emergency Medicine, Faculty of Medicine, University of Thessaly, Biopolis, 41500 Larissa, Greece; 2Medical School, European University of Cyprus, Nicosia 2404, Cyprus; magounakikelly@gmail.com; 3Department of Respiratory Medicine, Faculty of Medicine, University of Thessaly, Biopolis, 41500 Larissa, Greece; raniakotsiou@gmail.com (O.K.); eri.rouka@gmail.com (E.R.); kgourg@uth.gr (K.G.); 4ICU Department, Henry Dynant Hospital Center, 11526 Athens, Greece; perlykos@yahoo.com; 5Critical Care Department, “Sotiria” General Hospital of Chest Diseases, 11527 Athens, Greece; sotikaka@yahoo.com

**Keywords:** biomarkers, chronic obstructive pulmonary disease, exacerbations, lung aging, oxidative stress

## Abstract

Globally, chronic obstructive pulmonary disease (COPD) remains a major cause of morbidity and mortality, having a significant socioeconomic effect. Several molecular mechanisms have been related to COPD including chronic inflammation, telomere shortening, and epigenetic modifications. Nowadays, there is an increasing need for novel therapeutic approaches for the management of COPD. These treatment strategies should be based on finding the source of acute exacerbation of COPD episodes and estimating the patient’s own risk. The use of biomarkers and the measurement of their levels in conjunction with COPD exacerbation risk and disease prognosis is considered an encouraging approach. Many types of COPD biomarkers have been identified which include blood protein biomarkers, cellular biomarkers, and protease enzymes. They have been isolated from different sources including peripheral blood, sputum, bronchoalveolar fluid, exhaled air, and genetic material. However, there is still not an exclusive biomarker that is used for the evaluation of COPD but rather a combination of them, and this is attributed to disease complexity. In this review, we summarize the clinical significance of COPD-related biomarkers, their association with disease outcomes, and COPD patients’ management. Finally, we depict the various samples that are used for identifying and measuring these biomarkers.

## 1. Introduction

The pathogenesis of chronic obstructive pulmonary disease (COPD) involves a series of cellular and molecular processes driven by cytokines, chemokines, growth factors, oxidative stress, apoptosis, proteases-antiproteases imbalance, chronic tissue damage, and repair, and the relevant receptors and genetic signals [1]. It has become evident that COPD is not a single disease entity but comprises a set of distinct phenotypes with different underlying molecular and genetic pathways [2]. COPD-related research is increasingly focused on the search for biomarkers of the disease [3]. By definition, a biomarker is “objectively measured and evaluated as an indicator of normal biological processes, pathogenic processes, or pharmacologic responses to a therapeutic intervention” [4]. The multifactorial nature of the pathobiology of COPD implies that a large number of molecules could serve as biomarkers indicative of different aspects of the disease such as the presence or the extent of pulmonary damage, lung or systemic inflammation, and comorbidities. Moreover, there is a great interest in developing biomarkers that could enable the clear delineation and quantification of the distinct characteristics and outcomes associated with the various COPD phenotypes. The development of relevant biomarkers is also essential for the evaluation and discovery of individualized therapies that would combine improved clinical efficacy with minimal risk of adverse effects for patients in each of the COPD phenotypes.

Over the last years, a variety of biomarkers have been evaluated in COPD, derived from various sources including peripheral blood and genetic material [5]. Some of these tests have been reported to be useful to some degree for diagnostic or therapeutic purposes. However, in most cases, the value of the potential biomarkers in guiding COPD phenotyping and management is limited [6]. The Evaluation of COPD Longitudinally to Identify Predictive Surrogate End-points (ECLIPSE) study of 2.164 patients with COPD has provided valuable information concerning COPD phenotypes and relevant biomarkers and/or genetic parameters [7]. According to the ECLIPSE study, some potential biomarkers have been identified, but no single biomarker seems to fulfill all the necessary requirements. All these findings deserve further prospective validation in other COPD cohorts. Table 1 shows biomarkers currently being under investigation. This review aimed to present the most important recent findings on potential biomarkers derived from the peripheral blood and genetic material that can be used clinically to impact patient care in COPD.

## 2. Complete Blood Count-Based Biomarkers

The phenotype of frequent exacerbators (≥2 per year) is characterized by a persistent elevation of white blood cell (WBC) count, among other features [8,9]. These patients have a high risk of hospitalization and poorer prognosis, including increased mortality [10,11]. Additionally, peripheral eosinophil level from a complete blood count (CBC) has been evaluated as a surrogate marker for corticosteroid responsiveness, and eosinophilic bronchitis [12,13]. The analysis of the ECLIPSE cohort has shown that eosinophilic inflammation (eosinophil counts ≥2% at all visits) was present in 37.4% of patients and was associated with spirometrically and clinically less severe COPD [14]. More importantly, a higher blood eosinophil count appears to indicate a subgroup of COPD patients in which the use of inhaled corticosteroids (ICS) results in reduced exacerbation frequency [12,15,16]. Furthermore, significant elevations in peripheral eosinophil counts have been associated with a higher exacerbation rate when ICS is withdrawn [17,18,19]. In the setting of an acute exacerbation, peripheral eosinophil counts have been reported as an indicator of patients that would benefit from systemic corticosteroids [20]. This strategy could spare the use of corticosteroids in patients without eosinophilia avoiding possible adverse effects and poorer recovery rates [21,22]. The 2019 treatment guidelines from the Global Initiative for Chronic Obstructive Lung Disease (GOLD) have recommended blood eosinophil counts ≥300 cells·μL^−1^ in stable COPD as the diagnostic criterion for initiating therapy with ICS/long-acting β-agonist (LABA) [23]. However, further prospective validation is necessary to allow the widespread clinical implementation of peripheral blood eosinophils as a guide for the initiation of inhaled or systemic corticosteroids. The optimal cut-off value for the definition of significant eosinophilia has not been established yet [24]. Optimal cut-off values are fundamental for the clinical distinction of patients that would benefit from inhaled corticosteroid therapy. The realization that eosinophilic inflammation is significant in a subgroup of patients with COPD has been recently translated in the development of interleukin (IL)-5 receptor antagonists (benralizumab and mepolizumab) [25,26]. IL-5 expression is transduced through a cooperative signaling network that promotes eosinophil precursor maturation and prolongs the survival of eosinophils. Previously the inhibition of IL-5 has been proved efficacious in reducing severe exacerbations in patients with severe asthma [27]. In a recent phase III trial, mepolizumab at a dose of 100 mg was associated with a lower annual rate of moderate or severe exacerbations in patients with COPD and an eosinophilic phenotype documented by a blood eosinophil count of at least 150 per cubic millimeter at screening or at least 300 per cubic millimeter during the previous year [26]. Benralizumab is a human monoclonal antibody that enhances antibody-dependent cell-mediated cytotoxicity by the blockade of IL-5Rα expressed by eosinophils and basophils [28]. In COPD patients with higher baseline blood eosinophil counts the administration of benralizumab resulted in improved lung function and health status and a trend toward reduction in exacerbations [25]. Based on these encouraging findings phase III studies have been initiated.

Numerous serum biomarkers have been previously tested for their diagnostic, phenotyping, and prognostic ability in cohorts of COPD patients. Most of them are inflammatory markers, such as C-reactive protein (CRP) and IL-6, the circulating levels of which have been found elevated in patients with COPD [29,30,31]. These studies produced a series of associations most of which are summarized in Table 2. These findings should be taken into account as they may indicate clinical aspects useful for the integrative assessment of the COPD patient [32,33]. However, the reported associations are highly variable and sometimes poorly reproducible and seem inadequate to establish a clear relationship with relevant clinical outcomes of the disease. Thus, at present no single serum biomarker exhibits performance characteristics that allow a definite clinical translation and therapeutic guidance in COPD patients [34]. Therefore, the need for well-validated serum/plasma biomarkers in the COPD population remains.

COPD phenotype with persistent systemic inflammation has been proposed. This phenotype is associated with poor prognosis, including increased mortality [8,9]. The combination of a selective inflammatory panel with BODE index measured at baseline has been shown to improve the ability to predict 3-year and 8-year mortality [47]. This issue has been further addressed by the ECLIPSE study by analyzing the levels and the relationship of a panel of inflammatory markers [WBC count, CRP, IL-6, IL-8, fibrinogen, and tumor necrosis factor-alpha (TNF-α)] with clinical characteristics and relevant outcomes at 3 years follow-up [7,8]. According to this analysis, distinct inflammatory patterns seem to emerge. COPD patients show higher levels of some biomarkers (especially fibrinogen, and IL-6) in comparison with the control smokers and non-smokers. A subgroup of patients accounting for 16% of the sample had persistently high levels of inflammatory biomarkers and this was associated with a greater risk of exacerbations at 1 year regardless of the level of airflow limitation. Additionally, in smokers without COPD the levels of specific markers (IL-8, TNF-α) are increased compared with both patients with COPD and non-smokers without COPD [7,8]. Finally, plasma CRP, fibrinogen, serum TNFα levels, and immunoglobulin E (IgE) levels are higher in patients with asthma and COPD overlap (ACO) compared to those with COPD alone [8,48]. Such findings represent another indication that patients with ACO may share a specific inflammatory pattern more responsive to corticosteroids and perhaps with a different prognosis. The data concerning therapeutic efficacy in this phenotype are limited since randomized controlled clinical trials exclude asthmatic smokers and patients with possible ACO.

## 3. Oxidative Stress Biomarkers

Oxidative stress has a significant role in the pathophysiology of COPD. The existence of an essential equilibrium among the cellular oxidant and antioxidant mechanisms plays a crucial role in the preservation of the physiological function of the respiratory system. This disequilibrium of the oxidant-antioxidant mechanism is attributed to the increased oxidants and decreased antioxidants production which subsequently contributes to COPD severity [35,46]. Thus, oxidative stress has significant adverse effects like DNA and protein damage, and lipid destruction. So far, numerous oxidative stress biomarkers including both oxidants and antioxidants have been examined in COPD cases [36]. Several both non-invasive [exhaled breath condensate (EBC), sputum] and invasive methods [bronchoalveolar lavage (BAL), bronchoscopy] have been used for the detection of biomarkers in the COPD [49]. The most commonly used biological samples for the identification of biomarkers include the blood, sputum, BAL, and exhaled air [50]. Malondialdehyde (MDA) and thiobarbituric acid reactive substances (TBARS) are the universal biomarkers used for the detection of oxidative stress mostly in blood samples [36]. Studies compared the MDA values in blood samples of stable COPD cases and acute exacerbation of COPD (AECOPD) cases. The results revealed high MDA values in those with AECOPD [35]. Additionally, the studies assessed the levels of antioxidant biomarkers like glutathione (GSH), glutathione peroxidase (GSH-Px), and superoxide dismutase (SOD) and found them at low levels in the AECOPD patients [35]. Moreover, many studies investigated the concentration of dietary antioxidants especially vitamin A, E, and C, and found a remarkable reduction of their levels in the AECOPD [35]. Various oxidative stress biomarkers such as ethane can be detected at high levels in the exhaled air of COPD individuals. Of interest, ethane is also associated with COPD intensity [41]. Furthermore, studies examined the levels of the hydrogen peroxide (H2O2) biomarker which was remarkably elevated in EBC samples of COPD individuals [35]. High levels of 4-hydroxynonenal (4HNE), MDA, and 8-isoprostane have been detected in patients with either stable or AECOPD [46]. Notably, 8-isoprostane has been investigated in several pulmonary samples like exhaled air and sputum. Many studies found increased 8-isoprostane concentration in the sputum of COPD individuals and even higher in AECOPD. However, a further increase of 8-isoprostane was observed in the sputum of smokers compared to that of non-smokers, concluding that smoking is a significant confounding factor. Publication data revealed that smokers had elevated 8-isoprostane for a period of at least three months following smoking cessation [49]. Thus, the elevated levels of this compound in ex-smokers indicate an endogenous source of oxidative stress and simultaneously an ongoing pulmonary inflammation [41]. Similarly, MDA was studied in both sputum and EBC samples. EBC MDA levels were higher in AECOPD individuals compared to stable cases, but sputum MDA levels were further elevated in AECOPD compared to stable COPD cases. Interestingly, only one study has examined biomarkers in BAL [35]. GSH, the main antioxidant in the respiratory system, was investigated in BAL of both stable and AECOPD cases and was found to be decreased in severe AECOPD patients [35]. However, the results regarding blood GSH levels are still controversial [49].

Moreover, increased γ-glutamyltransferase (GGT), an enzyme that is involved in the pathway of GSH production and the development of many conditions characterized by oxidative stress, was detected in the plasma of COPD patients [36]. A positive link between GGT and CRP levels, as well as COPD severity, has been reported [35].

## 4. Age-Related Biomarkers

COPD is a chronic disease that is characterized by accelerated lung aging [51]. Several pathological mechanisms of accelerated lung aging have been examined in COPD patients including telomere attrition, epigenetic changes, stem cell exhaustion, cellular senescence, epigenetic changes, oxidative stress, mitochondrial dysfunction, and genomic instability. Cellular senescence describes a process in which the presence of stressors such as reactive oxygen species (ROS) leads cells to a permanent cell arrest state which is correlated with phenotypic alterations [52]. Cellular senescence is an established lung aging process that is associated with both functional and structural impairment in COPD patients [53]. Moreover, cellular senescence occurs in patients with emphysematous lungs and is correlated with shorter telomeres and reduced anti-aging molecules, indicating accelerated lung aging [52]. Additionally, studies have found increased levels of cellular senescence biomarkers like p16, p19, and p21, which are tumor suppressors and cyclin kinase inhibitors, in COPD individuals [53,54]. Sirtuin-1 (SIRT1) is another cellular senescence biomarker that, in COPD patients is expressed in low concentrations in the respiratory epithelium of the small airways [53]. Furthermore, smoking contributes to the high levels of senescence biomarkers like protein p21 and b-galactosidase [53,54].

Stem cell exhaustion is another significant mechanism of aging that contributes to the COPD pathogenesis [54]. Stem cells can replace and regenerate diseased cells, a property which is lost with age leading to age-related disorders [52,54]. In COPD patients the stem cells which are responsible for the regeneration of respiratory epithelium possess a diminished capacity of cellular regeneration which subsequently affects the entire cellular repair process [52]. Concerning the relationship between COPD and other physiological parameters, it was found that PaO_2_ has an important association with telomere length in COPD individuals because these patients have regular incidences of hypoxia, especially during COPD exacerbations, sleep, and physical activity which induces oxidative stress to the cells [54]. The length of telomeres has been used as a biomarker of aging and disease progression in COPD patients. Specifically, studies compared the length of the telomeres in leukocytes of COPD patients with both a control group and smokers without COPD. The results revealed shorter telomere length in white blood cells of COPD subjects even after matching for sex, age, and tobacco exposure [55,56]. However, shorter telomere length was observed in elderly patients with COPD and smokers without COPD. Moreover, a few large-scale studies have indicated a moderate association between telomere length and respiratory function related to forced expiratory volume in one second (FEV1) [57,58]. Recent studies in which many biomarkers of aging were examined in COPD patients, revealed that telomere length is the sole biomarker related to respiratory function. Additionally, a link between telomere length and other comorbidities like hypertension, cancer, diabetes mellitus in COPD subjects has not been established [56]. Systemic inflammation is another factor that affects the telomere length in COPD. High levels of inflammatory markers such as IL-6, IL-1β, IL-8, Transforming growth factor-beta (TGF-B) have been found in the respiratory tract and bloodstream of COPD subjects [56]. Specifically, IL-6 is considered a pro-inflammatory cytokine involved in aging and low-grade activation of chronic inflammation [54]. Interestingly, the telomeres length shortening was negatively associated with the levels of IL-6 in COPD individuals. Thus, inflammation influences the telomeres length [56]. Studies have also investigated the role of two main age-related hormonal biomarkers, dehydroepiandrosterone (DHEA) and growth hormone (GH) in the process of accelerated lung aging. The results revealed a remarkable decrease in DHEA and GH levels in COPD patients. Additionally, it was shown that a negative association exists among DHEA, GH, and age in COPD and non-COPD patients. Specifically, it was found that COPD patients present early biological aging, ranging from 13 to 23 years, compared to non-COPD patients depending on the variations of DHEA and GH levels. Moreover, both DHEA and GH showed an important association with many respiratory parameters like FEV1 and PaO2 [56]. To sum up, further studies are needed to elucidate the molecular mechanisms of aging in patients with aging-related diseases like COPD [56].

## 5. Bronchial Biomarkers

Several non-invasive techniques (exhaled air, induced sputum, EBC) and invasive diagnostic techniques (BAL, respiratory tissue biopsies) are currently used for the detection of bronchial biomarkers in COPD patients [42].

Regarding non-invasive methods, induced sputum is used for both the identification of inflammatory biomarkers and the presence of eosinophilia [42]. The sputum sampling is collected mostly from the large respiratory airways [45]. The main sputum inflammatory biomarkers that have been identified include cytokines especially IL-6, IL-8, and TNF-α. High levels of these cytokines were found in individuals with AECOPD. Moreover, it was noticed that in patients with COPD exacerbations the level of inflammatory biomarkers has an inverse relationship with FEV1 [43]. Additionally, high proteases levels, like matrix metalloproteinase (MMP)-8, MMP-9, MMP-12, and neutrophil elastase were found in sputum samples of COPD individuals. Also, elevated proportions of extracellular matrix (ECM) structural components were identified in the sputum of COPD patients [45]. Eosinophil peroxidase is used as an indicator of eosinophilia in the sputum samples of COPD patients and has been found elevated in COPD sputum samples [44]. In the exhaled air method, the fraction of exhaled nitric oxide (FeNO) was found as the predominant biomarker [42]. The measurement of FeNO concentration is considered difficult and can be influenced by factors like smoking and ICS. Remarkably, increased FeNO levels have been detected at the beginning of the AECOPD [43]. Moreover, FeNO has a positive association with FEV1 following COPD treatment, and specifically, the reduction of FeNO levels is associated with increased FEV1 [43].

BAL performed in COPD patients revealed the presence of numerous inflammatory biomarkers like myeloperoxidase, eosinophil cationic protein, and IL-8 at increased concentrations. Few studies showed increased levels of tryptase and histamine in COPD patients. Additionally, BAL sample results demonstrated high proteases and low anti-protease levels [41]. However, BAL has several limitations due to its invasive nature, sampling method, and other confounding factors like smoking and ICS [41]. Studies that have compared BAL samples of COPD patients versus healthy smokers, showed decreased concentrations of certain biomarkers including epidermal growth factor receptor (EGFR), human serum albumin (HSA), and alpha-1-antitrypsin (A1AT) in the BAL samples of COPD patients. In contrast, healthy smokers had elevated levels of HSA, A1AT, and MMP3. Simultaneously, BAL of COPD patients had high levels of IL-8, Tissue inhibitor matrix metalloproteinase 1 (TIMP1), and calprotectin indicative of respiratory tract neutrophilia. Interestingly, HSA and A1AT were the dominant biomarkers in BAL samples [37].

As mentioned above, oxidative stress has a dominant role in the pathogenesis of COPD, with adverse outcomes in the structural components of the cells. Inflammation of the respiratory system leads to ROS production by certain cells like macrophages, neutrophils, and epithelial cells. H2O2 and 8-isoprostaglandin F2a (8-isoprostane) are the two predominant biomarkers of oxidative stress that can be detected in an exhaled breath at a high concentration [42]. The use of EBC in COPD patients has shown high levels of both H2O2 and 8-isoprostaglandin F2a. Moreover, both biomarkers are associated with the intensity of the COPD [45]. The EBC pH is another potential biomarker, but it is still not well studied [43].

## 6. Mucine-Producing Pathways

The lining of the respiratory tract has an important mucociliary clearance mechanism which is involved in its protection by various environmental and infectious factors. Any impairment of this clearance mechanism can lead to mucus build-up and peripheral airways obstruction [59,60]. Increased mucin production is the main feature involved in COPD pathogenesis and has been associated with a high risk of morbidity, mortality, as well as COPD exacerbations and disease severity [61,62]. Numerous mucin genes have been identified in the respiratory system of humans. The most abundant are the Mucin (MUC)5AC and MUC5B [61]. These two mucin genes are responsible for both the composition of mucus and its movement along the airways [62]. Specifically, in the airways of healthy individuals, MUC5B is the predominant mucin gene that is related to mucus clearance from the airways. Opposed to that, the MUC5AC gene is expressed at low rates in the airways of healthy patients. Remarkably, a significant increase in the expression of MUC5AC has been related to inflammatory states and muco-obstructive lung diseases compared to MUC5B [59,61]. Furthermore, several other factors are implicated in the increased production of mucins such as oxidative stress and smoking. Indeed, smoking augments mostly the MUC5AC expression and to a slighter degree the expression of MUC5B, in the respiratory epithelium of smokers compared to the non-smoker’s [59,63]. The mechanism by which cigarette smoke enhances mucus production still needs further investigation, although oxidative stress is currently considered the main causative factor [63]. Moreover, studies showed a positive relationship between disease severity and high MUC5AC levels [59]. The results of pulmonary function tests revealed an inverse relationship among MUC5AC, and pulmonary function expressed as forced expiratory flow (FEF) 25–75%, which was not found for the MUC5B mucin gene [59,60]. Interestingly, a marked decrease in the pulmonary function (i.e., FEV1) with a parallel rise in MUC5AC levels was observed in smokers compared to ex-smokers with normal MUC5AC levels. Recent publications showed that ex-smokers with COPD had marginally increased MUC5AC levels which did not return to normal levels following smoking cessation. Similar reversibility rates were observed for the levels of MUC5B as well. However, early smoking cessation prior to airway obstruction was found to prevent pulmonary function decline and MUC5AC regulation. In conclusion, MUC5AC can be used as a potential biomarker for COPD detection, prognosis, and effectiveness of the treatment [59].

## 7. Extracellular Vesicles as Biomarkers in COPD

Recent studies have examined the use of extracellular vesicles (EVs) as both diagnostic and prognostic biomarkers in COPD and their potential role in distinguishing COPD exacerbations from the stable state as well as defining the COPD phenotype [64]. EVs are membrane particles that are released in systemic circulation by endothelial cells undergoing either apoptosis or activation [40]. Several body fluids have been used for the isolation of Evs including blood, urine, and BAL [64]. EVs express several endothelial cell markers which are specific to the stimuli that caused their release. More specifically, EVs that express CD31+ are related to apoptosis of endothelial cells whereas the expression of CD62E is related to activation of endothelial cells. Furthermore, the expression of CD51 is associated with the chronic injury [39]. Studies found that there is a link between CD31+ EVs and decreased diffusing capacity of carbon monoxide (DLCO). In addition, there is a negative association among EVs expressing CD31+ and FEV1. Increased levels of CD31+ EVs were found in COPD patients and were related to the severity of COPD. Additionally, CD31+ EVs were related to the emphysematous phenotype of COPD on imaging studies [39]. In contrast, EVs expressing CD62E+ were increased only in individuals with severe COPD and were related to lung hyperinflation [39]. Takahashi et al. showed that increased CD62E+ EVs (E-selectin) were observed in COPD individuals with regular episodes of AECOPD and those prone to exacerbations [40]. Concerning CD51+ EVs, elevated levels were detected in COPD individuals. However, no connection was observed between CD51+ or CD62E+ EVs and DLCO [39]. Overall, further research is needed upon the use of EVs as biomarkers for the diagnosis and management of COPD in the clinical practice [39].

## 8. Genetic Biomarkers

Mutations in the SERPINA1 gene leading to a1-antithrypsin deficiency represent at present the only established genetically based phenotype of COPD for which targeted therapy exists. Although this genetic condition accounts only for 1–2% of the total COPD population it may respond positively to replacement therapy with an alpha 1 proteinase inhibitor [65,66]. Increased susceptibility to smoking-induced emphysema has been associated with polymorphisms of the heme oxygenase (HO-1) promoter leading to reduced HO-1 expression [67,68]. Additionally, susceptibility to emphysema has been recently linked to a variant (single nucleotide polymorphisms; SNP) of the BICD1 gene [69]. Patients with ACO are characterized by an enhanced expression of several genes, such as toll-like receptor 10 (TLR10) which has been previously implicated in the pathogenesis of the asthma [48]. Other studies on genetic polymorphisms have identified several genes associated with the pathogenesis of different characteristics of COPD: cholinergic nicotine receptor alpha 3/5 (CHRNA3/5), iron regulatory binding protein 2 (IREB2), hedgehog-interacting protein (HHIP), family with sequence similarity 13, member A (FAM13A), and advanced glycosylation end product-specific receptor (AGER) [70,71,72,73,74,75]. Previously reported associations of these genetic variants include: airflow limitation (CHRNA3/5, IREB2, HHIP), [76] emphysema susceptibility and severity (CHRNA3/5, BICD1), [77] chronic bronchitis phenotype, [78] exacerbation rate (HHIP) [79] and pulmonary hypertension pathogenesis [80]. Nevertheless, the majority of these findings comprise suggestive liaisons and their clinical translation and applicability require replication in additional studies [81].

## 9. COPD Exacerbation-Related Biomarkers

AECOPD is related to poor health outcomes, increased morbidity, and mortality rates [82,83]. The diagnostic and therapeutic management of AECOPD is still considered insufficient due to its heterogeneity and complexity [84]. Currently, the diagnostic approach of AECOPD is mostly based on the patient’s symptomatology [83,84]. Thus, biomarkers should be actively investigated to incorporate them in the clinical assessment [83].

Nowadays, inflammatory biomarkers serve as diagnostic and prognostic tools in patients with AECOPD. Such inflammatory biomarkers that are commonly used include CRP, procalcitonin (PCT), and fibrinogen. Additionally, biomarkers like serum amyloid A (SAA), serum surfactant protein-D (SP-D), vascular endothelial growth factor (VEGF), troponin-T (TNT), 4-HNE, β-thromboglobulin, platelet factor-4 (PF4), and copeptin were proven beneficial for the assessment of intensity and outcome of AECOPD. Recently, FeNO was identified as a promising upcoming biomarker [84].

The results of the ECLIPSE study indicated that AECOPD is associated with increased levels of inflammatory biomarkers including white blood cells, CRP, and fibrinogen during the first year of follow-up [10,85]. Interestingly, no link has been observed among elevated levels of IL-6 and acute COPD exacerbations [86]. However, three studies found an increased level of IL-6 in AECOPD, although the statistical significance was not reported in [87]. The TNF-α biomarker was also found to be elevated during AECOPD [87]. Recent studies found a positive interdependence between high levels of fibrinogen, CRP, and leukocytes in patients presenting with COPD exacerbations [38,85,88]. Remarkably, the ECLIPSE, COPDGene, and the Copenhagen Lung Study found that an increased number of eosinophils in the blood were associated with COPD exacerbations [83]. Moreover, the COPDGene cohort study demonstrated a significant association between serious AECOPD and five biomarkers, the plasminogen activator inhibitor-1 (PAI-1), soluble receptor for advanced glycation end products (sRAGE), A1AT, brain-derived neurotrophic factor (BDNF), and C-X-C Motif Chemokine Ligand 5 (CXCL5).

Other biomarkers that have been identified in prior studies include CRP, leukocytes, eosinophils, ILs, fibrinogen, extracellular adenosine triphosphate (eATP), and extracellular heat shock protein 70 (eHsp70) [89]. The extracellular ATP (eATP), a key molecule of the pro-inflammatory cascade pathway associated with respiratory airway diseases, was investigated in the plasma of COPD patients. High levels of eATP were associated with the frequency of COPD exacerbations, symptoms severity, and rate of airflow decline [90]. Similar results were reported for the heat shock protein 70, an important pro-inflammatory molecule having a crucial role in the regulation of immunological pathways [89,90].

Other studies have examined the connection between the AECOPD and respiratory tissue destruction. The results showed an increased number of circulating structural proteins of the respiratory extracellular matrix which can be used as diagnostic biomarkers [83,91,92]. In 2017, Noell et al. showed that the combined set of elevated CRP and neutrophil levels in conjunction with dyspnea can accurately diagnose AECOPD [84]. Additionally, studies have evaluated the measurement of volatile organic compounds during expiration in AECOPD cases and considered it as a useful, non-invasive diagnostic tool [83,93].

## 10. Combination of Biomarkers

Numerous protein biomarkers have been studied in the bloodstream of COPD individuals aiming to assess the disease outcomes [94]. Specifically, studies have found a link between biomarkers like the SP-D, CRP, fibrinogen, and high mortality rates in COPD patients. Nevertheless, no association has been found among SP-D, CRP, fibrinogen, soluble receptor of activated glycogen end-product (sRAGE), club cell protein 16 (CC-16), and the degree of FEV1 decline, hospitalizations, and COPD exacerbations [91].

The COPDGene cohort study showed that both the decrease of lung function and the progression of emphysema could be most reliably estimated by measuring the level of a specific panel of biomarkers (sRAGE, CC-16, and fibrinogen). Similarly, the ECLIPSE study used the same panel of biomarkers combined with SP-D and CRP strengthening, even more, the estimations [94]. Furthermore, elevated levels of the IL-1α, IL-1β, IL-6, IL-8, TNF-α cytokines have been found in the serum and sputum of COPD subjects. A positive association was found between CRP and IL-1β in the blood of COPD patients. Moreover, IL-1β and IL-6 were negatively associated with FEV1. Additionally, a positive correlation was observed between the cytokines IL-1β, IL-6, TNF-α, and COPD severity [89].

To sum up, more investigations are needed to find the association between the aforementioned panels of biomarkers and the COPD outcomes [91].

## 11. Conclusions

Age-related diseases like COPD are increasing in frequency due to population aging. In the last years, numerous biomarkers have been investigated in COPD patients, although their significance is not well established. The study of appropriate and ideal biomarkers is of high importance for the disease diagnosis, prognosis, and treatment effectiveness. Moreover, COPD prognosis and response to treatment could be assessed by evaluating the combination of biomarkers in COPD individuals. Many types of biological samples and diagnostic techniques are used for the detection of these numerous biomarkers in COPD patients, with each one having its sensitivity.

The recent therapeutic methods for COPD are mostly targeting the patients’ COPD-related symptoms. For this reason, further research is warranted to develop novel therapies which could target the underlying pathways that lead to COPD pathogenesis. Furthermore, the disease heterogeneity among COPD individuals especially at the level of COPD severity, progression, and patients’ comorbidities as well as clinical status, could set the foundations for more personalized management of these patients. Specifically, the measurement and evaluation of each patients’ unique biomarker panel could be a quite convenient approach in the upcoming years. As a result of this, more effective, and targeted therapies could be followed.

## Figures and Tables

**Table 1 jpm-12-00379-t001:** Biomarkers under investigation for COPD management.

Specimen Type	Readily Available and Currently Used Biomarkers	Extensively Investigated Biomarkers but Not Sufficiently Validated	Less Investigated Biomarkers
Peripheral Blood (plasma/ serum)	Eosinophils	MDA	Vitamins A, E, and C
	CRP	GSH, GSH-Px, SOD	GGT
		IL-6, TNFα, MCP-1	vWF
			Extracellular vesicles (CD62E+, CD31+)
Exhaled air	FeNO		Ethane
Sputum		IL-6, IL-8, TNF-α	8-isoprostane
		MPO	MDA
		MMP-8, MMP-9, MMP-12, neutrophil elastase, Eosinophil peroxidase	SOD, GSH-Px
			Leptin
Exhaled breath condensate		8-isoprostane	MDA
		H_2_O_2_	IL-8
Bronchoalveolar lavage fluid		Glutathione	EGFR, HSA, A1AT, TIMP1, IL-8 and Cal-protectin
Urine			8-isoprostane

MDA: Malondialdehyde, GSH: glutathione, GSH-Px: glutathione peroxidase, SOD: superoxide dismutase, GGT: γ-glutamyltransferase, CRP: C-reactive protein, IL: interleukin, TNF-α: tumor necrosis factor-alpha, MCP-1: monocyte chemoattractant protein-1, vWF: von Willebrand factor, FeNO: fraction of exhaled nitric oxide, MPO: myeloperoxidase, MMP: matrix metalloproteinase, EGFR: epidermal growth factor receptor, HSA: human serum albumin, A1AT: alpha-1-antitrypsin, TIMP1: tissue inhibitor matrix metalloproteinase 1.

**Table 2 jpm-12-00379-t002:** Summary table of biomarkers in COPD management.

Specimen Type	Biomarker	Main Findings	First Author [Ref]
Peripheral Blood (plasma/ serum)	MDA	MDA levels were significantly higher in patients with AECOPD compared to stable COPD	Zinellu E. [35]
	Vitamins A, E, and C	Levels of vitamins A and E, but not C were significantly lower in patients with AECOPD than stable COPD	
	GSH, GSH-Px, SOD	Decreased levels of these antioxidant biomarkers were found in the plasma of patients with AECOPD compared to stable COPD	
	GGT	GGT levels were significantly higher in patients with AECOPD (adjusted for age, gender, smoking status) compared to stable COPD and a positive association was reported with CRP	Zinellu E. [36]
	IL-6, TNFα, MCP-1 vWF	Elevated serum levels of IL-6, TNFα and MCP-1, depict the systemic inflammation that occurs in COPD patients Increased concentration of vWF was reported in the serum of COPD smokers	Röpcke S. [37]
	CRP	Positive association of CRP with morbidity, mortality, and frequency of exacerbations Negative association with lung function parameters	Röpcke S. [37]
		CRP was used for the confirmation of AECOPD	Lacoma A. [38]
		CRP was used as a prognostic biomarker and as a marker of inflammatory response in COPD patients Increased CRP levels were found in both patients with AECOPD and stable COPD CRP had a sensitivity of 72.5% and a specificity of 100% for the diagnosis of patients with AECOPD	Heidari B. [31]
	Extracellular vesicles	CD31+ EVs, suggestive of endothelial cell apoptosis, were elevated in patients with emphysema CD62E+ EVs indicative of endothelial activation were elevated in severe COPD and hyperinflation	Thomashow M.A. [39]
		Higher baseline CD62E+ EVs may indicate COPD patients who are susceptible to exacerbation	Takahashi T [40]
	Blood eosinophilia	Peripheral blood eosinophilia (above 0.2 × 10^9^/L) can be used for the detection of sputum eosinophilia mostly in stable COPD It is considered a sensitive biomarker for the detection of sputum eosinophilia in AECOPD (sensitivity 90%, specificity 60%)	Negewo N.A. [13]
Exhaled air	Ethane	Elevated levels of ethane are found in exhaled air of COPD patients and are associated with COPD severity	Barnes P.J. [41]
	FeNO	Smoking is considered a significant limitation of FeNO use because it negatively affects its concentration FeNO is elevated in patients with asthma-like component of COPD Potential biomarker for estimating treatment response in COPD patients	Angelis N. [42]
		FeNO levels increased at the onset of AECOPD and decreased with resolution FeNO had an inverse relationship with FEV1% Increase of FEV1% following a decrease in FeNO (sensitivity 74%, specificity 75%)	Koutsokera A. [43]
Sputum	MPO, 8-isoprostane	No significant elevation of MPO and 8-isoprostane was found in patients with AECOPD	Zinellu E. [35]
		Increased levels of 8-isoprostane were detected in COPD patients compared to non-smokers and smokers without COPD A positive association was observed between 8-isoprostane and pulmonary function parameters	Comandini A. [44]
	MDA, SOD, GSH-Px	Elevated levels of MDA, and reduced SOD and GSH-Px were observed in the sputum of patients with AECOPD compared to stable COPD A positive association was detected among these biomarkers in induced sputum	Zinellu E. [35]
	MMP-8, MMP-9, MMP-12, neutrophil elastase, Eosinophil peroxidase	Elevated levels of these biomarkers were found in COPD patients	Barnes P.J. [45]
			Comandini A. [44]
	IL-6, IL-8, TNF-α, Leptin	Elevated levels of IL-6, IL-8, TNF-a were observed in severe COPD cases compared to less severe COPD Increased levels of IL-8 were associated with COPD severity (predicted FEV1%) progression and AECOPD	Barnes P.J. [45]
	IL-6, IL-8, TNF-α	Elevated levels of IL-6, IL-8 and TNF-α are observed in patients with AECOPD compared to stable COPD	Koutsokera A. [43]
Exhaled breath condensate	MDA, H_2_O_2_	No difference was observed in the MDA levels in the EBC of patients with AECOPD and stable COPD H_2_O_2_ was highly elevated in both patients with AECOPD and stable COPD	Zinellu E. [35]
		MDA was elevated in the EBC of COPD patients and was even higher in patients with an AECOPD Elevated levels of H_2_O_2_ were found in both patients with AECOPD and stable COPD	Barnes P.J. [41]
	8-isoprostane	Increased levels of 8-isoprostane were observed in COPD patients	Chamitava L. [46]
			Koutsokera A. [43]
	8-isoprostaglandin F2a (8-isoprostane)	8-isoprostane was associated with disease severity Its concentration was found to be higher in COPD patients compared to smokers without COPD	Barnes P.J. [45]
	IL-8	There is an inverse relationship of IL-8 and PFTs at the onset of an AECOPD	Koutsokera A. [43]
Bronchoalveolar lavage fluid	Glutathione	Reduced glutathione levels were observed in severe AECOPD compared to stable COPD	Zinellu E. [35]
		Lower levels of glutathione were observed in frequent AECOPD compared to stable COPD.	Barnes P.J. [41]
	EGF-R, HSA, A1AT, TIMP1, IL-8 and Calprotectin	Low levels of EGF-R, HSA and A1AT were found in the BAL of COPD patients Increased concentrations of TIMP1, IL-8 and Calprotectin were detected in the BAL of COPD patients that were correlated with airway inflammation	Röpcke S. [37]
Urine	8-isoprostane	Increased levels of 8-isoprostane were observed in the urine of COPD patients	Chamitava L. [46]

MDA: Malondialdehyde, AECOPD: acute exacerbation of COPD, GSH: glutathione, GSH-Px: glutathione peroxidase, SOD: superoxide dismutase, GGT: γ-glutamyltransferase, CRP: C-reactive protein, IL: interleukin, TNF-α: tumor necrosis factor-alpha, MCP-1: monocyte chemoattractant protein-1, vWF: von Willebrand factor, FeNO: fraction of exhaled nitric oxide, FEV1: forced expiratory volume in one second, MPO: myeloperoxidase, MMP: matrix metalloproteinase, EBC: exhaled breath condensate, PFTs: pulmonary function tests, EGFR: epidermal growth factor receptor, HSA: human serum albumin, A1AT: alpha-1-antitrypsin, BAL: bronchoalveolar lavage, TIMP1: tissue inhibitor matrix metalloproteinase 1, EVs: extracellular vesicles.

## Data Availability

Not applicable.

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
