# Peer review of "Incorporating Biomarkers in COPD Management: The Research Keeps Going"

_jpm, 2022, doi:10.3390/jpm12030379_

Round 1

Reviewer 1 Report

Row 140 – it is stated:  ...“ that patients with ACOS may share...“ ACOS is no more valid abbreviation, as it was recognized that overlap between asthma and COPD  is not a separate syndrome, then it is just an Asthma and COPD overlap (ACO), two diseases with the some shared characteristics. It should be corrected.

The same about ACO in row 143.

Very systematically, comprehensive, and didactically written overview on current knowledge on biomarkers in different body compartments in COPD, during a stable phase as well as in acute exacerbation of COPD. I like it a lot.

Author Response

Detailed response to reviewers’ comments

Reviewer(s)' Comments to Author:

Reviewer: 1

“Row 140 – it is stated:  ...“ that patients with ACOS may share...“ ACOS is no more valid abbreviation, as it was recognized that overlap between asthma and COPD  is not a separate syndrome, then it is just an Asthma and COPD overlap (ACO), two diseases with the some shared characteristics. It should be corrected.”

Thank you very much for the comment. The requested change has been made.

“The same about ACO in row 143.”

The requested change has been made throughout the manuscript.

“Very systematically, comprehensive, and didactically written overview on current knowledge on biomarkers in different body compartments in COPD, during a stable phase as well as in acute exacerbation of COPD. I like it a lot.”

We would like to thank the distinguished reviewer for his/her favorable comments and the time he/she devoted in reviewing our manuscript.

Reviewer 2 Report

The manuscript is a review on biomarkers of COPD. The topic is of interest, even though to date minimal data are available concerning a significant role of biomarkers in clinical practice. The manuscript is, for the most part, well written.

I have, however, a few concerns that I would like to share with the Authors

  • Biomarkers are in general presented as a mere list, without any effort to structure the information in a way that might make reading the manuscript more appealing and the information more useful. The Authors could have tried, for example, to separate biomarkers that are readily available and currently used (e.g. peripheral blood eosinophils to inform ICS therapy), from those that have been extensively investigated but not sufficiently validated to be incorporated into clinical practice (e.g. procalcitonin to inform antibiotic therapy in AECOPD), from those that are likely to be abandoned due to negative results of confirmatory studies and from those that have been started to be investigated only relatively recently. Again, the above example is just one of many ways that might be attempted to make the reported information more useful than an unstructured list of molecules.
  • There is no mention to extracellular vesicles. The role of EV as biomarkers in COPD has been extensively investigated (see, for example, Takahashi et al, Thorax 2012 and BMJ open 2014; Thomashow et al, AJRCCM 2013). A paragraph dedicated to these structures might be relevant.

A few other specific issues should be addressed before the manuscript is accepted for publication.

  • Page 7; lines 221-224. The two sentences “there was no remarkable correlation …” and “… a moderate association” appear contradictory
  • Paragraph 8 and 9. All in all, these paragraphs appear more confusing than the remaining sections of the manuscript. More specifically:
  • Page 9, line 347. The statement that biomarkers should be included in the clinical assessment is not accurate. The point is that biomarkers should be actively investigated in order to incorporate them in the clinical assessment. In its present form, the sentence hints at an inappropriate behavior of physicians that fail to do what should be done.
  • Page 9, line 350. The statement that such biomarkers (the sentence includes SP-D and SAA) are commonly used is not accurate.
  • Page 10, lines 366-372. Please read the text carefully: A1-AT is mentioned twice (lines 366 and 369); the definition of C-C motif CC16 and ligand 18 is confusing (“chemokine (C-C motif), CC16, SP-D, and ligand 18”)
  • Page 10, line 384-385: CRP, SRAGE and fibrinogen are not cytokines.
  • Page 10, lines 391 and 395: again, there seems to be a contradiction between the sentence “IL1, IL1B [please note the repetition] (…) showed no relationship” and “however, IL1B was negatively associated”
  • Page 10, line 398-401. The description of eATP and HSP70 does not belong to the paragraph “multiple biomarkers”.
  • Minor point: throughout the paragraph, Greek letters have been substituted with Latin characters (e.g. TNF-a, IL1-B)

Author Response

Detailed response to reviewers’ comments

Reviewer(s)' Comments to Author:

Reviewer: 2

“The manuscript is a review on biomarkers of COPD. The topic is of interest, even though to date minimal data are available concerning a significant role of biomarkers in clinical practice. The manuscript is, for the most part, well written.”

Thank you very much for the time you devoted in reviewing this manuscript. We hope that the revised manuscript will satisfy your high standards and you will finally consent to its publication.

“I have, however, a few concerns that I would like to share with the Authors. Biomarkers are in general presented as a mere list, without any effort to structure the information in a way that might make reading the manuscript more appealing and the information more useful. The Authors could have tried, for example, to separate biomarkers that are readily available and currently used (e.g. peripheral blood eosinophils to inform ICS therapy), from those that have been extensively investigated but not sufficiently validated to be incorporated into clinical practice (e.g. procalcitonin to inform antibiotic therapy in AECOPD), from those that are likely to be abandoned due to negative results of confirmatory studies and from those that have been started to be investigated only relatively recently. Again, the above example is just one of many ways that might be attempted to make the reported information more useful than an unstructured list of molecules.”

Thank you very much for your insightful comment. Our aim was to provide an update on current biomarkers used in COPD management according to their origin for example, complete blood count-based biomarkers, oxidative stress biomarkers, age-related biomarkers, etc. in order to make it easier for the reader to find and become updated on the specific biomarkers that he/she is interested in. On the other hand, in table 2, we have presented biomarkers according to the specimen type; peripheral blood, exhaled air, sputum, exhaled breath condensate, BAL. We agree that separating biomarkers according to their availability and validation is useful, thus we have inserted a new table (new table 1) providing this information.

“There is no mention to extracellular vesicles. The role of EV as biomarkers in COPD has been extensively investigated (see, for example, Takahashi et al, Thorax 2012 and BMJ open 2014; Thomashow et al, AJRCCM 2013). A paragraph dedicated to these structures might be relevant.”

Thank you very much for the comment. The requested addition has been made.

“A few other specific issues should be addressed before the manuscript is accepted for publication. Page 7; lines 221-224. The two sentences “there was no remarkable correlation …” and “… a moderate association” appear contradictory”

Thank you very much for the comment. The first sentence has been deleted and it is now clear that “a few large-scale studies have indicated a moderate association among telomere length and respiratory function related to forced expiratory volume in one second (FEV1)”.

“Paragraph 8 and 9. All in all, these paragraphs appear more confusing than the remaining sections of the manuscript. More specifically: Page 9, line 347. The statement that biomarkers should be included in the clinical assessment is not accurate. The point is that biomarkers should be actively investigated in order to incorporate them in the clinical assessment. In its present form, the sentence hints at an inappropriate behavior of physicians that fail to do what should be done.”

The requested change has been made.

“Page 9, line 350. The statement that such biomarkers (the sentence includes SP-D and SAA) are commonly used is not accurate.”

Thank you very much for your comment. The sentence has been corrected.  

“Page 10, lines 366-372. Please read the text carefully: A1-AT is mentioned twice (lines 366 and 369); the definition of C-C motif CC16 and ligand 18 is confusing (“chemokine (C-C motif), CC16, SP-D, and ligand 18”)”

Thank you very much for the comment. All required corrections have been made.

“Page 10, line 384-385: CRP, SRAGE and fibrinogen are not cytokines.”

Thank you very much for the comment. All necessary corrections have been made.

“Page 10, lines 391 and 395: again, there seems to be a contradiction between the sentence “IL1, IL1B [please note the repetition] (…) showed no relationship” and “however, IL1B was negatively associated””

Thank you very much for the comment. All required changes have been made.

“Page 10, line 398-401. The description of eATP and HSP70 does not belong to the paragraph “multiple biomarkers”.”

Thank you very much for the comment. The description of eATP and HSP70 has been moved to section 8 “COPD Exacerbation-Related Biomarkers”.

“Minor point: throughout the paragraph, Greek letters have been substituted with Latin characters (e.g. TNF-a, IL1-B)”

The requested changes have been made.

Round 2

Reviewer 2 Report

The Authors have extensively reviewed the manuscript. It is my opinion that the manuscript is now acceptable in its present form.